# Critical roles of mTOR Complex 1 and 2 for T follicular helper cell differentiation and germinal center responses

Jialong Yang[1], Xingguang Lin[1], Yun Pan[1], Jinli Wang[2], Pengcheng Chen[2], Hongxiang Huang[1], Hai-Hui Xue[3], Jimin Gao[2], Xiao-Ping Zhong[1,4,5]*

[1]Department of Pediatrics, Division of Allergy and Immunology, Duke University Medical Center, Durham, United States; [2]School of Laboratory Medicine, Wenzhou Medical University, Wenzhou, China; [3]Department of Microbiology, Carver College of Medicine, University of Iowa, Iowa, United States; [4]Department of Immunology, Duke University Medical Center, Durham, United States; [5]Hematologic Malignancies and Cellular Therapies Program, Duke Cancer Institute, Duke University Medical Center, Durham, United States

**Abstract** T follicular helper (Tfh) cells play critical roles for germinal center responses and effective humoral immunity. We report here that mTOR in CD4 T cells is essential for Tfh differentiation. In *Mtor^{f/f}-Cd4Cre* mice, both constitutive and inducible Tfh differentiation is severely impaired, leading to defective germinal center B cell formation and antibody production. Moreover, both mTORC1 and mTORC2 contribute to Tfh and GC B cell development but may do so via distinct mechanisms. mTORC1 mainly promotes CD4 T cell proliferation to reach the cell divisions necessary for Tfh differentiation, while Rictor/mTORC2 regulates Tfh differentiation by promoting Akt activation and TCF1 expression without grossly influencing T cell proliferation. Together, our results reveal crucial but distinct roles for mTORC1 and mTORC2 in CD4 T cells during Tfh differentiation and germinal center responses.

*For correspondence: xiaoping. zhong@duke.edu

## Introduction

T follicular helper (Tfh) cells belong to a special subset of CD4 T cells that are essential for germinal center (GC) formation, Ig-class switch and hypermutation, memory B cell and long-lived plasma cells generation, and establishment of long-term protective immunity (*King, 2009*; *Tangye et al., 2013*; *Craft, 2012*; *Crotty, 2014*; *Vinuesa et al., 2016*). Differentiation of Tfh cells is considered a multi-stage and tightly regulated process (*Lu et al., 2011*). After engagement with antigenic peptide-MHC complexes, which dendritic cells present through their T cell receptors (TCRs) and subsequent activation, a portion of CD4 T cells upregulate ICOS, Bcl-6, and CXCR5 expression, migrate to the T cell–B cell border in the spleen and lymph nodes (LNs) (*Kerfoot et al., 2011*; *Fazilleau et al., 2009*; *Suan et al., 2015*). Engagement of T cells with matured B cells via CD40-CD40L and ICOS-ICOSL interactions promotes GC-responses (*Liu et al., 2015*; *Pratama et al., 2015*; *Awe et al., 2015*). Subsequently, the differentiating Tfh cells migrate into B cell follicles and further differentiate into GC-Tfh cells to direct generation of GC B cells. GC-Tfh cells produce cytokines such as IL-21 and IL-4 to regulate Ig-class switch and hypermutation in B cells (*Schmitt et al., 2009*).

Recent studies have identified multiple signaling mechanisms that control Tfh cell differentiation. TCR, costimulatory, and cytokine signals are known to contribute to Tfh differentiation at different stages (*Vinuesa et al., 2016*). TCR signal initiates T cell activation, and its duration and strength shapes Tfh differentiation and function (*Deenick et al., 2010*; *Baumjohann et al., 2013*;

*Fazilleau et al., 2009*). ICOS functions as a costimulatory receptor, signaling through the PI3K-Akt cascade to promote transcription of Bcl-6 (7, 17, 18), a transcription factor crucial for Tfh differentiation (*Johnston et al., 2009*; *Nurieva et al., 2009*; *Gigoux et al., 2009*; *Yu et al., 2009*; *Wang et al., 2014*). Akt phosphorylates Foxo1, leading to its sequestration in the cytosol and subsequent release of its suppression of Bcl-6 transcription (*Stone et al., 2015*). Additionally, ICOS increases cMAF expression, which in turn promotes expression of IL-21 (21), a cytokine critical for Tfh generation and GC-formation (*Nurieva et al., 2008*; *Vogelzang et al., 2008*). TCF1 and LEF1 transcription factors regulate the expression of multiple genes such as ASCL2, Blimp1-Bcl-6 axis, IL-6Rα, gp130, and ICOS that are involved in Tfh differentiation (*Choi et al., 2015*; *Liu et al., 2014*; *Xu et al., 2015*). β-catenin is a known coactivator of TCF1 and LEF1, and induced β-catenin activation has been shown to promote ASCL expression (*Liu et al., 2014*). Glycogen synthase kinase 3β (GSK3β) phosphorylates β-catenin, after which β-catenin is targeted for degradation (*Staal et al., 2008*; *Xue and Zhao, 2012*), and Akt can phosphorylate and inactivate GSK3β (*Hur and Zhou, 2010*). Thus, in addition to regulating Foxo1, Akt could potentially regulate GSK3β/β-catenin/TCF1 to shape Tfh differentiation. IL-2 signals to activate STAT5, leading to the inhibition of Tfh differentiation (*Johnston et al., 2012*).

The mammalian/mechanistic target of rapamycin (mTOR) is a serine/threonine kinase that can integrate multiple signals to control cell growth, proliferation, differentiation, metabolism, and survival (*Laplante and Sabatini, 2012*). mTOR signals mainly through two distinct complexes, mTORC1 and mTORC2, which are defined, respectively, by the signature components regulatory-associated protein of mTOR (Raptor) and rapamycin-insensitive companion of mTOR (Rictor). mTORC1 phosphorylates multiple substrates such as ribosomal protein S6 kinase (S6K) and eIF4E-binding protein 1 (4EBP1) to promote ribosome biogenesis, cap-dependent translation, and lipid and nucleic acid synthesis. mTORC2 phosphorylates completely different effector molecules such as PKCα, glucocorticoid-induced protein kinase 1 (SGK1), and Akt at serine 473 residue to regulate cytoskeleton reorganization, metabolism, and survival. In T cells, the PI3K/Akt, RasGRP1/Ras/Erk, and PKCθ-CARMA1 pathways are critical for TCR-induced mTORC1 and mTORC2 activation (*Gorentla et al., 2011*; *Hamilton et al., 2014*). mTOR and its tight regulation are pivotal for proper T cell activation (*Chi, 2012*; *O'Brien et al., 2012*; *O'Brien et al., 2011*; *Wu et al., 2011*; *Yang et al., 2011*), Th1/2/17 differentiation (*Delgoffe et al., 2011*; *Lee et al., 2010*; *Zhang et al., 2012*), effector/memory CD8 T cell differentiation (*Araki et al., 2009*; *Rao et al., 2010*), Treg function (*Park et al., 2013*; *Zeng et al., 2013*), and iNKT maturation and effector lineage differentiation (*Wu et al., 2014*; *Wu et al., 2014*; *Shin et al., 2014*; *Wei et al., 2014*; *Yang et al., 2015*; *Prevot et al., 2015*; *Zhang et al., 2014*), and extrinsic control of intrathymic αβ T and γδ T cell development (*Wang et al., 2016*; *Wang et al., 2016*).

The role of mTOR in Tfh differentiation is largely unclear. A recent report suggests that mTORC1 inhibits Tfh differentiation based on data from rapamycin inhibition and shRNA knockdown (*Ray et al., 2015*). However, another study has found that a hypomorphic mutation of mTOR causes reduced Tfh responses (*Ramiscal et al., 2015*). While these studies support a role of mTOR in Tfh responses, they do not distinguish between the roles of mTORC1 and mTORC2 or rule out potential effects from other immune cell lineages. Using mice selectively deficient of mTOR, Raptor/mTORC1, and Rictor/mTORC2 in T cells, we report here that mTOR deficiency caused severe decreases in constitutive Tfh and GC-B cells in the mesenteric lymph nodes (mLNs) and Peyer's patches (PPs), correlated with drastic decreases in virtually all serum IgG subtypes in unchallenged naïve mice. Moreover, mTOR-deficient CD4 T cells fail to differentiate to Tfh cells following antigen immunization, resulting in impaired GC-B cell formation and antigen-specific IgG responses. Both mTORC1 and mTORC2 contribute significantly to Tfh differentiation but appear to do so via distinct mechanisms. mTORC1-deficient CD4 T cells are defective in proliferation and cannot reach divisions that Tfh differentiation may require. In contrast, mTORC2-deficient CD4 T cells do not display obvious impairment in proliferation; instead, they display impaired Akt activation because of decreased phosphorylation at S473, subsequently increased cell death, impaired GSK3β phosphorylation at S9 and inactivation, and decreased β-catenin and TCF1 expression. Importantly, overexpression of a phosphomimetic mutant of Akt, Akt S473D, or TCF1 can partially restore Tfh differentiation, suggesting an mTORC2/Akt/TCF1 axis for Tfh differentiation.

## Results

### Deficiency of mTOR in T cells impaired constitutive Tfh differentiation and GC B cell responses

Because oral antigens and gut-microbiota-derived antigens stimulate T and B cells, the PPs contain constitutive GCs. About one-third of PP CD4 T cells from $Mtor^{f/f}$(WT) mice were CXCR5$^+$PD-1$^+$ Tfh

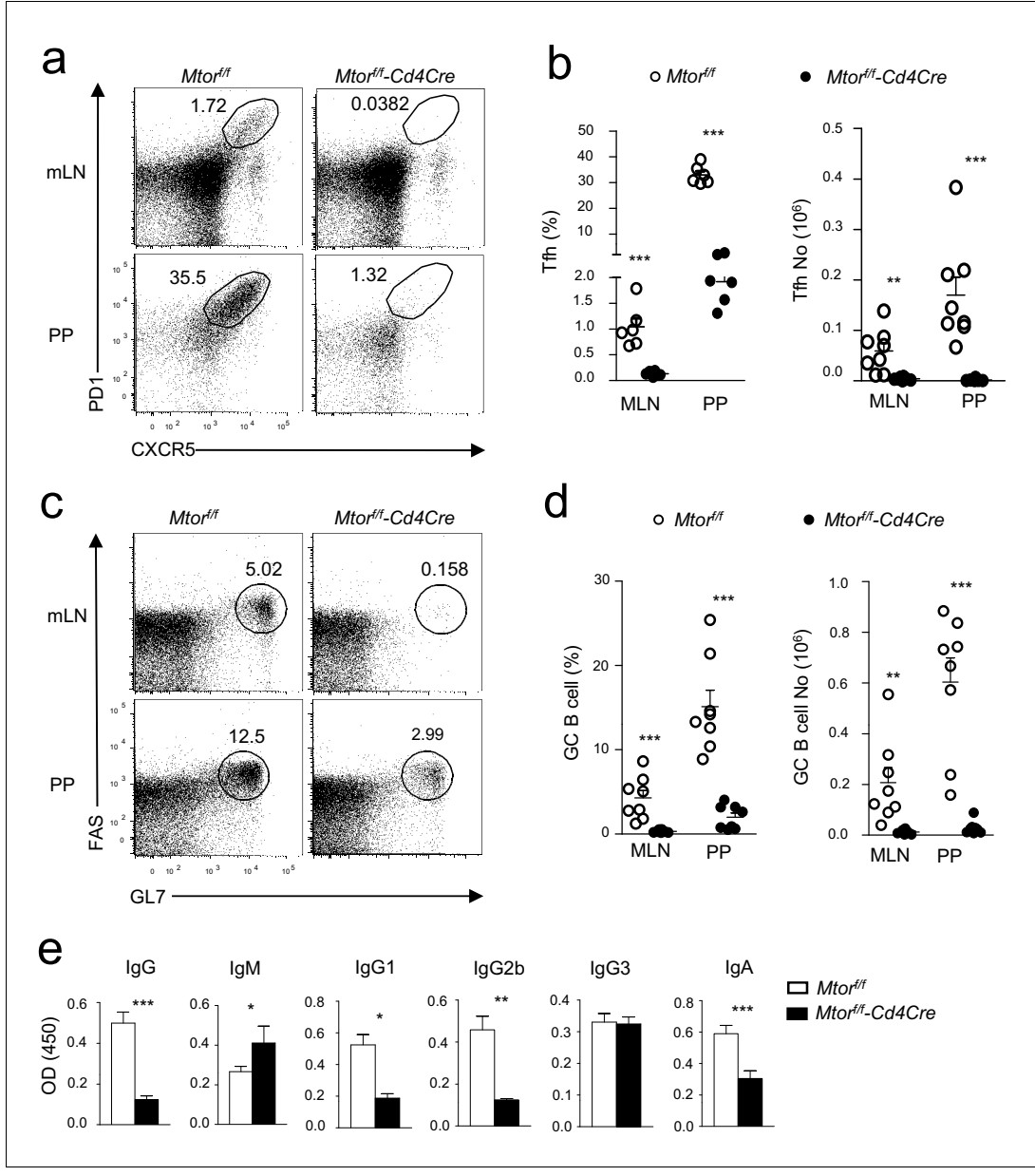

**Figure 1.** Critical role of mTOR for constitutive Tfh and GC responses. We collected sera, mLNs, and PPs from 2–3-month-old $Mtor^{f/f}$ and $Mtor^{f/f}$-$Cd4Cre$ for analysis. (**a**) Representative dot-plots of CXCR5 and PD1 staining in gated CD4$^+$TCRβ$^+$ T-cells from mLNs and PPs. (**b**) Scatter plots represent mean ± SEM of Tfh percentages (left panel) and numbers (right panel). (**c**) Representative dot-plots show GL7 and Fas staining in gated CD93$^-$B220$^+$IgM$^-$IgD$^-$ B cells from mLNs and PPs. (**d**) Scatter plots represent mean ± SEM of GC-B cell percentages (left panel) and numbers (right panel). (**e**) Relative serum IgM, IgG, and IgG subtypes (n ≥ 5) and fecal IgA (n = 19) levels measured by ELISA. Data represent or are calculated from at least five experiments (**a–d**) or two experiments (**e**). *p<0.05; **p<0.01; ***p<0.001 determined by unpaired two-tailed Student $t$-test.

cells (*Figure 1a and b*), and about 10% to 25% B220$^+$CD93$^-$IgM$^-$IgD$^-$ B cells were GL7$^+$Fas$^+$ GC B-cells (*Figure 1c and d*). Tfh and GC-B cells could also be detected in mLNs but at lower frequencies than PPs in WT mice. Both percentages (*Figure 1a and b*) and absolute numbers (right panel, *Figure 1b*) of Tfh cells substantially decreased in mLNs and PPs from *Mtor$^{f/f}$-Cd4Cre* mice compared with their littermate controls. Concordantly, GC B-cell percentages (*Figure 1c and d*) and absolute numbers (right panel, *Figure 1d*) in *Mtor$^{f/f}$-Cd4Cre* mLNs and PPs also dramatically decreased, correlated with reduced serum IgG but increased serum IgM levels (*Figure 1e*). Interestingly, IgG1, and IgG2b but not IgG3 levels decreased in mTOR deficiency mice, suggesting that the IgG3 class-switch occurred independently of mTOR signaling in CD4 T cells. Additionally, IgA secreted in the intestinal lumen decreased (*Figure 1e*), which was consistent with impaired GC-responses in PPs. Thus, mTOR deficiency in T cells severely compromised constitutive Tfh and GC responses in PPs and mLNs as well as overall humoral immunity.

## Contribution of mTORC1 and mTORC2 to constitutive Tfh and GC B cell responses

To further investigate the contribution of mTORC1 and mTORC2 to constitutive Tfh and GC B cell responses, we examined *Rptor$^{f/f}$-Cd4Cre* or *Rictor$^{f/f}$-Cd4Cre* mice and their littermate controls in a manner similar to that described in the previous section. Both *Rptor$^{f/f}$-Cd4Cre* (*Figure 2a,b*) and *Rictor$^{f/f}$-Cd4Cre* mice (*Figure 2c,d*) contained fewer Tfh cells in mLNs and PPs compared to their respective controls. To rule out the possibility that defective Tfh differentiation of *Rptor$^{f/f}$-Cd4Cre* T cells resulted from abnormal T cell development after *Rptor* deletion in developing thymocytes, we adoptively transferred a mixture of CD45.1 WT and CD45.2 *Rptor$^{f/f}$-Rosa26$^{CreER}$* CD4 T cells into Rag2 deficient mice. Recipients were injected with tamoxifen on 7, 8, and 11 days after reconstitution, then were examined on day 14. CXCR5$^+$PD1$^+$ Tfh cell percentages within CD45.1$^+$ WT and CD45.2$^+$*Rptor$^{f/f}$-Rosa26$^{CreER}$* CD4 T cells were similar in recipients without tamoxifen injection. However, in tamoxifen-treated recipients, CXCR5$^+$PD1$^+$ Tfh cell percentages in *Rptor$^{f/f}$-Rosa26$^{CreER}$* CD4 T cells were obviously decreased compared with WT controls in the same recipients or with *Rptor$^{f/f}$-Rosa26$^{CreER}$* CD4 T cells in mice without tamoxifen injection (*Figure 2—figure supplement 1*), further supporting the importance of mTORC1 for Tfh differentiation.

Coinciding with reduced Tfh cells, GC B-cells decreased in mLNs and PPs in both *Rptor$^{f/f}$-Cd4Cre* (*Figure 2e,f*) and *Rictor$^{f/f}$-Cd4Cre* mice (*Figure 2g,h*), although at magnitudes less severe than in *Mtor$^{f/f}$-Cd4Cre* mice. Moreover, total serum IgG but not IgM levels in *Rptor$^{f/f}$-Cd4Cre* (*Figure 2i*) and *Rictor$^{f/f}$-Cd4Cre* mice (*Figure 2j*) also decreased compared with controls. Interestingly, mTORC1 deficiency caused reduced IgG1 and IgG2b levels without obviously affecting IgG3 (*Figure 2i*), while mTORC2 deficiency resulted in decreases in IgG1, IgG2b, and IgG3 levels in serum (*Figure 2j*). However, unlike mTOR-deficient mice, neither *Rptor$^{f/f}$-Cd4Cre* nor *Rictor$^{f/f}$-Cd4Cre* mice had reduced fecal IgA levels (*Figure 2i,j*). Together, these observations suggested that both mTORC1 and mTORC2 contributed to constitutive Tfh and GC responses in mLNs and PPs and may synergistically or redundantly promote intestinal IgA responses.

## Effects of mTORC1 and mTORC2 deficiency on regulatory T cells

Regulatory T cells (Tregs) actively suppress immune responses and are essential for maintaining self-tolerance. It has been reported that mTOR deficiency causes relative enrichment of Tregs over conventional CD4 T cells (Tcon) (*Delgoffe et al., 2009*) and that mTORC1 but not mTORC2 is critical for Treg suppressive function (*Zeng et al., 2013*). In *Rptor$^{f/f}$-Cd4Cre* mice, CD4$^+$Foxp3$^+$ Treg percentages as well as Treg to Tcon ratios were similar to WT controls in the spleen, mLN, and PPs (*Figure 3a–c*). Treg numbers in *Rptor$^{f/f}$-Cd4Cre* mice were not obviously altered in the spleen and mLNs, but they were decreased in the PPs compared with WT controls (*Figure 3d*). The decrease of total PP cell numbers (14.17 ± 3.22 million in *Rptor$^{f/f}$* vs 6.13 ± 0.72 million in *Rptor$^{f/f}$-Cd4Cre*, p<0.05) was a major contributing factor for decreased PP Treg numbers in *Rptor$^{f/f}$-Cd4Cre* mice. In *Rictor$^{f/f}$-Cd4Cre* mice, Treg percentages and numbers were decreased in the spleen and PPs, but they were not significantly decreased in mLNs (*Figure 3g,h and j*), correlated with decreased total T cell numbers in the spleen and PPs in *Rictor$^{f/f}$-Cd4Cre* mice (data not shown). However, Treg to Tcon ratios were not significantly skewed (*Figure 3i*). It has been recently reported that T follicular regulatory cells (Tfr) play important roles in suppressing GC responses (*Sage and Sharpe, 2015*).

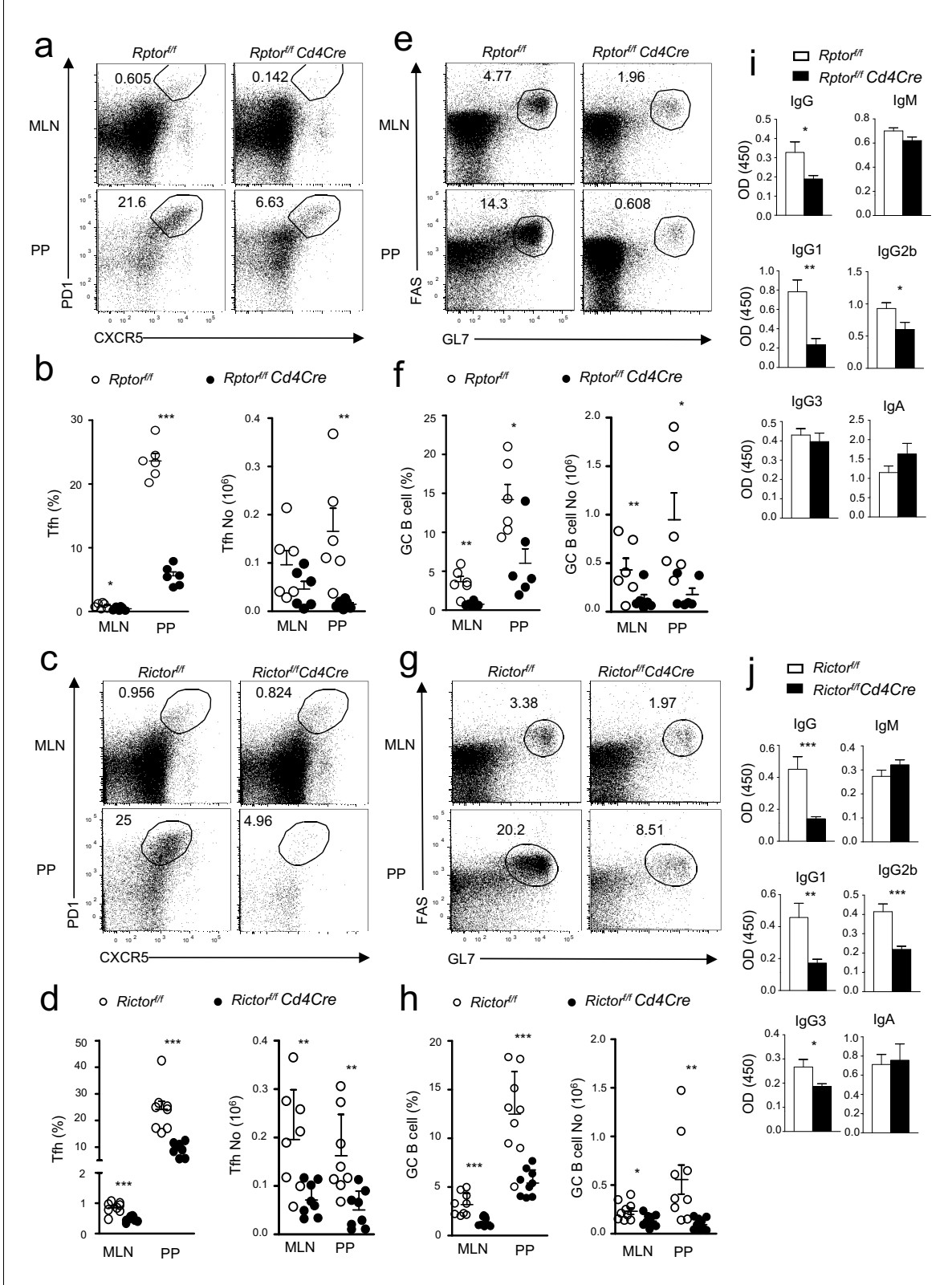

**Figure 2.** Contribution of mTORC1 and mTORC2 to the constitutive Tfh and GC B cell responses. We assessed *Rptor^f/f-Cd4Cre*, *Rictor^f/f-Cd4Cre*, and their littermates control mice using the procedure in *Figure 1*. (a, c) Representative dot-plots showing frequencies of Tfh cells in gated mLN and PP CD4^+TCRβ^+ T-cells from *Rptor^f/f-Cd4Cre* (a), *Rictor^f/f-Cd4Cre* (c) and their control mice. (b, d) Scatter plots representing mean ± SEM of mLN and PP Tfh percentages (left panel) and numbers (right panel) in *Rptor^f/f-Cd4Cre* (b), *Rictor^f/f-Cd4Cre* (d) and their control mice. (e, g) Representative dot-plots

*Figure 2 continued on next page*

*Figure 2 continued*

showing frequencies of GC B-cells population in gated CD93⁻B220⁺IgM⁻IgD⁻ B cells from mLNs and PPs of *Rptor^f/f-Cd4Cre* (e), *Rictor^f/f-Cd4Cre* (g) and their control mice. (f, h) Scatter plots representing mean ± SEM of GC-B cell percentages (left panel) and numbers (right panel) of *Rptor^f/f-Cd4Cre* (f), *Rictor^f/f-Cd4Cre*, (h) and their control mice. (i, j) Serum IgM, IgG, IgG subtypes, and fecal IgA levels from *Rptor^f/f-Cd4Cre* (i, n = 7), *Rictor^f/f-Cd4Cre* (j, n = 8), and their control mice measured by ELISA. Data shown represent or are calculated from at least five experiments (a–h) or two experiments (i, j). *p<0.05; **p<0.01; ***p<0.001 determined by two-tailed Student *t*-test.

The following figure supplement is available for figure 2:

**Figure supplement 1.** Deletion of Raptor in naïve CD4 T cells impaired constitutive Tfh differentiation.

The percentages of the Foxp3⁺ population within PP Tfh cells were decreased and Foxp3⁺ Treg numbers were more severely decreased in *Rptor^f/f-Cd4Cre* mice, suggesting that mTORC1 deficiency had a stronger effect on Tfr development/homeostasis than on conventional Tregs (*Figure 3e,f*). In *Rictor^f/f-Cd4Cre* mice, PP Tfr percentages within Tfh cells were not decreased, but their numbers were decreased due to reduced Tfh numbers (*Figure 3k,l*). Together, our data suggest that deficiency of either mTORC1 or mTORC2 did not increase the relative abundance of Tregs within CD4 T cells in the spleen, mLNs, and PPs or the Tfr ratios within Tregs in the PPs. Together with decreased total Treg and Tfr numbers in *Rptor* or *Rictor* deficient PPs, these data suggest that the decreased constitutive GC-B cell response was likely not caused by change of Treg or Tfr numbers in these mice. However, whether mTORC1 and mTORC2 may play a role in Tfr function remains to be determined.

## Impaired inducible Tfh and GC B cell development in the absence of mTOR in T cells

To examine the role of mTOR in Tfh and GC responses during immune responses, we immunized *Mtor^f/f-Cd4Cre* mice and their littermate controls with a T-cell-dependent antigen, NP17-CGG, in alum. On day 21 after immunization, fewer splenic Tfh cells were present in *Mtor^f/f-Cd4Cre* mice than in WT controls in both percentages and numbers (*Figure 4a,b*) with concomitantly reduced GC B-cells (*Figure 4c,d*). Induction of Bcl-6, a transcription factor crucial for GC B cell differentiation (*Fukuda et al., 1997*), was blunted in splenic and mLN B cells from *Mtor^f/f-Cd4Cre* mice after immunization (*Figure 4e*). Moreover, both total (NIP26) and high-affinity (NIP7) serum NIP-specific IgM and IgG levels were much lower in *Mtor^f/f-Cd4Cre* mice than in WT mice 7, 14, and 21 days after immunization (*Figure 4f*). Thus, mTOR in CD4 T cells was essential for inducing Tfh differentiation and GC B cell formation following immunization and for T-cell-dependent antigen-induced humoral immune responses.

## Critical roles for mTORC1 and mTORC2 in Tfh differentiation following antigen immunization

To determine the role of mTORC1 and mTORC2 in antigen-induced Tfh/GC-B cell responses, we immunized *Rptor^f/f-Cd4Cre*, *Rictor^f/f-Cd4Cre* mice, and their respective control mice with NP17-CGG in alum. Both *Rptor^f/f-Cd4Cre* (*Figure 5a,b*) and *Rictor^f/f-Cd4Cre* mice (*Figure 5c,d*) contained fewer splenic Tfh cells than did WT controls, correlated with decreased numbers of splenic GC B cells in *Rptor^f/f-Cd4Cre* (*Figure 5e,f*) and *Rictor^f/f-Cd4Cre* (*Figure 5g,h*) mice. Moreover, NIP-specific total and high-affinity IgG and IgM antibody levels decreased in immunized *Rptor^f/f-Cd4Cre* deficient mice (*Figure 5i*), indicating impaired humoral immune responses. In comparison with *Rptor*-deficient mice, neither total nor high-affinity NIP-specific IgM levels decreased in *Rictor^f/f-Cd4Cre* mice (*Figure 5j*). While *Rictor^f/f-Cd4Cre* mice manifested a trend of mildly decreased NIP specific total and high-affinity IgG levels, only decreases of total IgG on day 7 and high-affinity IgG on day 14 were statistically significant (*Figure 5j*). Because NIP-specific antibody responses in *Rictor*-deficient mice were not as severe as in *Rptor*-deficient mice, we further examined GC formation in *Rictor^f/f-Cd4Cre* mice on day 14 after immunizing them via immunofluorescence microcopy using PNA and Thy1.2 to detect GC B cells and T cells, respectively. As shown in *Figure 5k,l*, GC numbers were obviously less and sizes of individual GCs were noticeably smaller in *Rictor^f/f-Cd4Cre* spleens than in controls, confirming impaired GC responses in the absence of mTORC2 in T cells. Together,

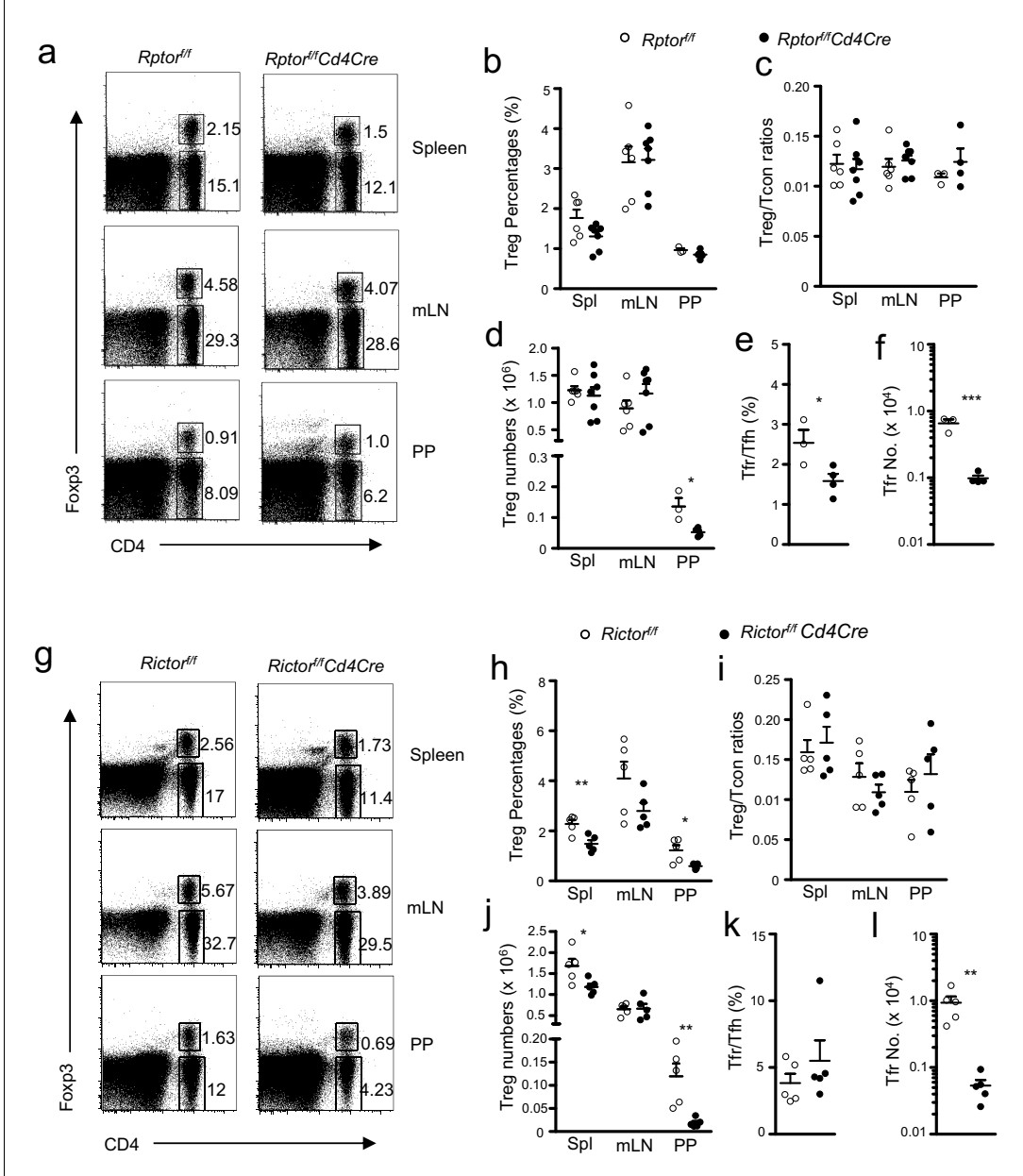

**Figure 3.** Effects of mTORC1 and mTORC2 deficiency on Tregs. We assessed *Rptor^f/f-Cd4Cre* (a–f), *Rictor^f/f-Cd4Cre* (g–l), and their littermate control mice using the procedure in *Figure 1* with the addition of intracellular Foxp3 staining. (a, g) Representative dot-plots showing Foxp3 and CD4 expression in lived gated splenocytes, mLN cells, and CD45$^+$ PP cells from *Rptor^f/f-Cd4Cre* (a), *Rictor^f/f-Cd4Cre* (g) and their control mice. (b, h) Scatter plots representing mean ± SEM of Treg percentages in *Rptor^f/f-Cd4Cre* (b), *Rictor^f/f-Cd4Cre* (h) and their control mice. (c, i) Scatter plots representing mean ± SEM of Treg/Tcon ratios in *Rptor^f/f-Cd4Cre* (c), *Rictor^f/f-Cd4Cre* (i) and their control mice. (d, j) Scatter plots representing mean ± SEM of Treg numbers in *Rptor^f/f-Cd4Cre* (d), *Rictor^f/f-Cd4Cre* (j) and their control mice. (e, k) Scatter plots representing mean ± SEM of Foxp3$^+$ Tfr percentages within total Tregs in PPs in *Rptor^f/f-Cd4Cre* (e), *Rictor^f/f-Cd4Cre* (k) and their control mice. (f, l) Scatter plots representing mean ± SEM of Foxp3$^+$ Tfr numbers within total Tregs in PPs in *Rptor^f/f-Cd4Cre* (f), *Rictor^f/f-Cd4Cre* (l) and their control mice. Data shown represent or are calculated from at least three experiments. *$p<0.05$; **$p<0.01$; ***$p<0.001$ determined by two-tailed Student *t*-test.

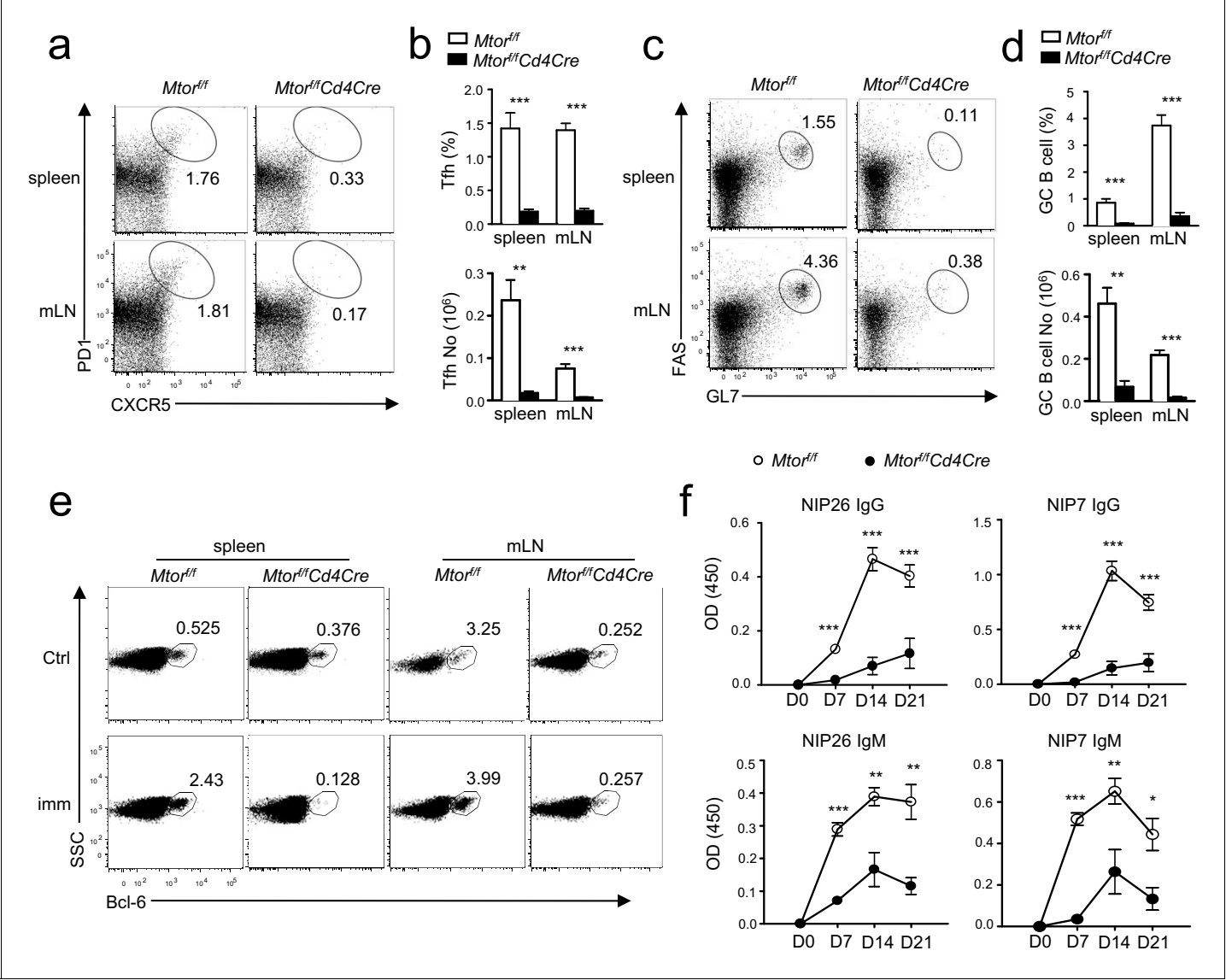

**Figure 4.** Deficiency of mTOR impaired inducible Tfh and GC B-cell responses following immunization. We injected *Mtor^f/f^* and *Mtor^f/f^-Cd4Cre* mice *intraperitoneally (i.p.)* with NP$_{17}$-CGG absorbed to alum. We collected sera the day before immunization and 7, 14, and 21 days after immunization, harvesting spleens and mLNs for analysis on day 21 after immunization. (**a**) Representative dot-plots showing CXCR5 and PD1 staining in gated splenic and mLN CD4$^+$TCRβ$^+$ T cells. (**b**) Bar graphs show mean ± SEM of Tfh percentages (top panel) and absolute numbers (bottom panel; WT, n = 9; KO, n = 7). (**c**) Representative dot-plots showing Fas and GL7 staining in gated B220$^+$CD93$^-$IgM$^-$IgD$^-$ B cells. (**d**) Bar graphs represent mean ± SEM of GC-B cell percentages (top panel) and absolute numbers (bottom panel; WT, n = 9; KO, n = 7). (**e**) Dot-plots show Bcl-6 expression in gated B220$^+$CD93$^-$IgM$^-$IgD$^-$ B cells from unimmunized mice or mice 21 days after immunization. (**f**) Serum NIP-specific IgG and IgM levels on indicated days after immunization detected by ELISA with NIP7- or NIP26-BSA coated plates (WT, n = 7; KO, n = 4). Data shown represent three (**f**) or are calculated from three (**a–e**) experiments. *p<0.05; **p<0.01; ***p<0.001 determined by two-tailed unpaired Student *t*-test.

both mTORC1 and mTORC2 contributed to Tfh/GC B cell responses, in which mTORC1 appeared to play a more important role than mTORC2.

## mTORC1 and mTORC2 intrinsically promote Tfh differentiation

Because Raptor and Rictor were absent in both CD4 and CD8 T cells starting in CD4$^+$CD8$^+$ thymocytes, potential preexisting abnormalities in *Rptor^f/f^*- or *Rictor^f/f^-Cd4Cre* mice could affect naïve CD4 T cell differentiation to Tfh cells. To provide further evidence that mTORC1 and mTORC2 intrinsically

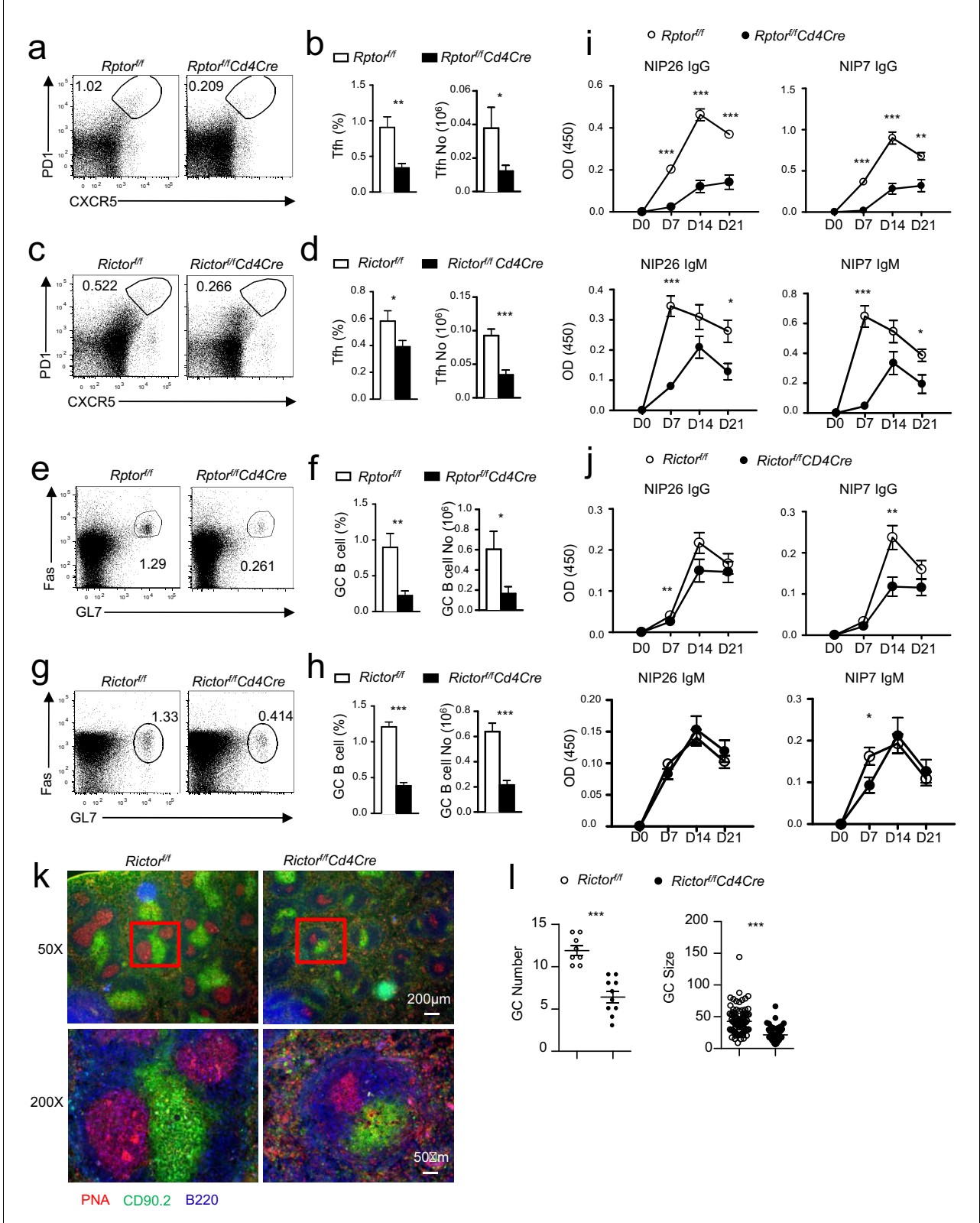

**Figure 5.** Contribution of mTORC1 and mTORC2 to inducible Tfh and GC B-cell responses. As in *Figure 4*, we immunized and examined *Rptor$^{f/f}$-Cd4Cre*, *Rictor$^{f/f}$-Cd4Cre* and their control mice with NP$_{17}$-CGG absorbed to alum. (a, c) Representative dot-plots showing PD1 and CXCR5 staining in gated splenic CD4$^+$ TCRβ$^+$ T cells from *Rptor$^{f/f}$-Cd4Cre* (a), *Rictor$^{f/f}$-Cd4Cre* (c) and their control mice. (b, d) Percentages and absolute numbers of splenic Tfh cells in *Rptor$^{f/f}$-Cd4Cre* (b), *Rictor$^{f/f}$-Cd4Cre* (d) and their control mice 21 days after immunization. (e, g) Representative dot-plots showing

*Figure 5 continued on next page*

*Figure 5 continued*

GL7 and Fas staining in gated splenic B220+CD93-IgM-IgD- B cells from *Rptor^f/f^-Cd4Cre* (e), *Rictor^f/f^-Cd4Cre* (g), and their control mice. (f, h) Percentages and absolute numbers of splenic GC B cells in *Rptor^f/f^-Cd4Cre* (f), *Rictor^f/f^-Cd4Cre* (h), and their control mice 21 days after immunization. (i, j) Relative serum NIP-specific IgG and IgM levels from *Rptor^f/f^-Cd4Cre* (i; n = 6) and *Rictor^f/f^-Cd4Cre* (j; n = 9) indicated days after immunization detected by ELISA with NIP7- or NIP26-BSA coated plates. (k, l) Impaired GC formation in Rictor-deficient mice. 14 days after immunization, we stained spleen thin sections from *Rictor^f/f^-Cd4Cre* and *Rictor^f/f^* mice with PNA, B220, and Thy1.2. Representative immunofluorescence images are shown (k; 50x and 200 x). Scatter plots depict mean ± SEM of GC numbers per view (l. left; 8 total views for WT and 10 views for KO) and sizes of the PNC+Thy1.2- GCs (l. right). Data shown represent or are calculated from three experiments. *p<0.05; **p<0.01; ***p<0.001 determined by two-tailed unpaired Student *t*-test.

control Tfh differentiation, we generated *Rptor^f/f^-Cd4Cre-OT2* and *Rictor^f/f^-Cd4Cre-OT2* mice as well as their respective control mice. The OT2 TCR-transgenic mice express the TCRVα2Vβ5 transgene that recognizes an ovalbumin peptide amino acid spanning amino acids from 323 to 339 (OVA$_{323-339}$) presented by H2-A$^b$ (*Barnden et al., 1998*). *Rptor^f/f^-* or *Rictor^f/f^-Cd4Cre* OT2 T cells showed similar CD62L+CD44- naïve, CD62L+CD44+ CM, and CD62L-CD44+ EM ratios to those of their respective WT controls (*Figure 6—figure supplement 1a*). *Rptor^f/f^-Cd4Cre* naïve OT2 T cells expressed similar CD25, CD122, CD69, and CD44 but decreased CD127 (IL7Rα) compared to its WT control (*Figure 6—figure supplement 1b*). *Rictor^f/f^-Cd4Cre* naïve OT2 T cells displayed similar CD25, CD69, CD122 and CD127 but noticeably decreased CD44 expression compared to their WT controls (*Figure 6—figure supplement 1c*).

We adoptively transferred naïve CD45.2+CD4+TCRVα2+ OT2 T cells from *Rptor^f/f^-Cd4Cre* OT2 (*Figure 6a–e*) or Thy1.1+ *Rictor^f/f^-Cd4Cre* OT2 (*Figure 6f–j*) transgenic mice and their corresponding WT OT2 control mice into congenic CD45.1+CD45.2+ or Thy1.1+Thy1.2+ WT recipients. We immunized recipient mice with OVA$_{323-339}$ peptide emulsified in the complete Freund's adjuvant (CFA) one day after they received OT2 T cells. On day 7 after immunization, CD45.1-CD45.2+*Rptor^f/f^-Cd4Cre* (*Figure 6a,b*) and Thy1.1-Thy1.2+ *Rictor^f/f^-Cd4Cre* (*Figure 6f,g*) donor derived OT2 cells were noticeably low in percentages and numbers within the gated CD4+TCRVα2+or CD4+TCRVβ2+ - OT2 population in draining LNs (dLNs) and spleen. Moreover, *Rptor* (*Figure 6a* bottom panel, *6c*)- or *Rictor* (*Figure 6f* bottom panel, *6g*)-deficient donor OT2 T cells contained much lower percentages of CXCR5+PD1+ Tfh cells than did their WT controls, leading to even more drastic decreases in total donor-derived Tfh cell numbers in dLNs and spleen in recipients with either *Rptor^f/f^-Cd4Cre* OT2 (*Figure 6d*) or *Rictor^f/f^-Cd4Cre* OT2 T cells (*Figure 6i*). Thus, data from adoptive transfer experiments demonstrated that naïve CD4+ T cells intrinsically required both mTORC1 and mTORC2 to differentiate into Tfh cells.

One potential mechanism that contributes to decreased Tfh responses of *Rptor* or *Rictor* deficient CD4+ T cells was increased death. *Rptor* deficient OT2 T cells, including both Tfh and non-Tfh populations, had similar percentages of dead cells compared with WT controls (*Figure 6e*). In contrast, *Rictor* deficient OT2 T cells showed increased death rates (*Figure 6j*). Thus, Rictor but not Raptor deficiency led to increased death of CD4+ T cells during immune responses.

## mTORC1 and mTORC2 promote Tfh differentiation via distinct mechanisms

To understand whether the impaired Tfh differentiation of *Rptor*- or *Rictor-deficient* CD4 T cells was associated with impaired proliferation, we adoptively transferred CFSE-labeled WT and *Rptor^f/f^-Cd4Cre*-OT2 cells into WT recipients and followed with OVA$_{323-339}$ peptide immunization. WT OT2 T cells demonstrated vigorous proliferation following immunization. In the absence of Raptor, OT2 T cells displayed an obvious defect in proliferation 3, 5, and 7 days after immunization (*Figure 7a*). Similarly, proliferation of Raptor-deficient OT2 T cells was defective following in vitro TCR stimulation (*Figure 7—figure supplement 1a*) without severe defects in upregulation of T cell activation markers CD69 and CD44 with the exception of decreased CD25 upregulation (*Figure 7—figure supplement 1b*), which was consistent with the role of mTORC1 for cell cycle entry after T cell activation (*Yang et al., 2013*). CXCR5+ or PD1+ Tfh cells, differentiated from WT naïve CD4 T cells, began to appear after at least six divisions (*Figure 7b–e*). While delayed in proliferative expansion, some Raptor-deficient OT2 cells eventually reached more than six divisions 7 days after immunization.

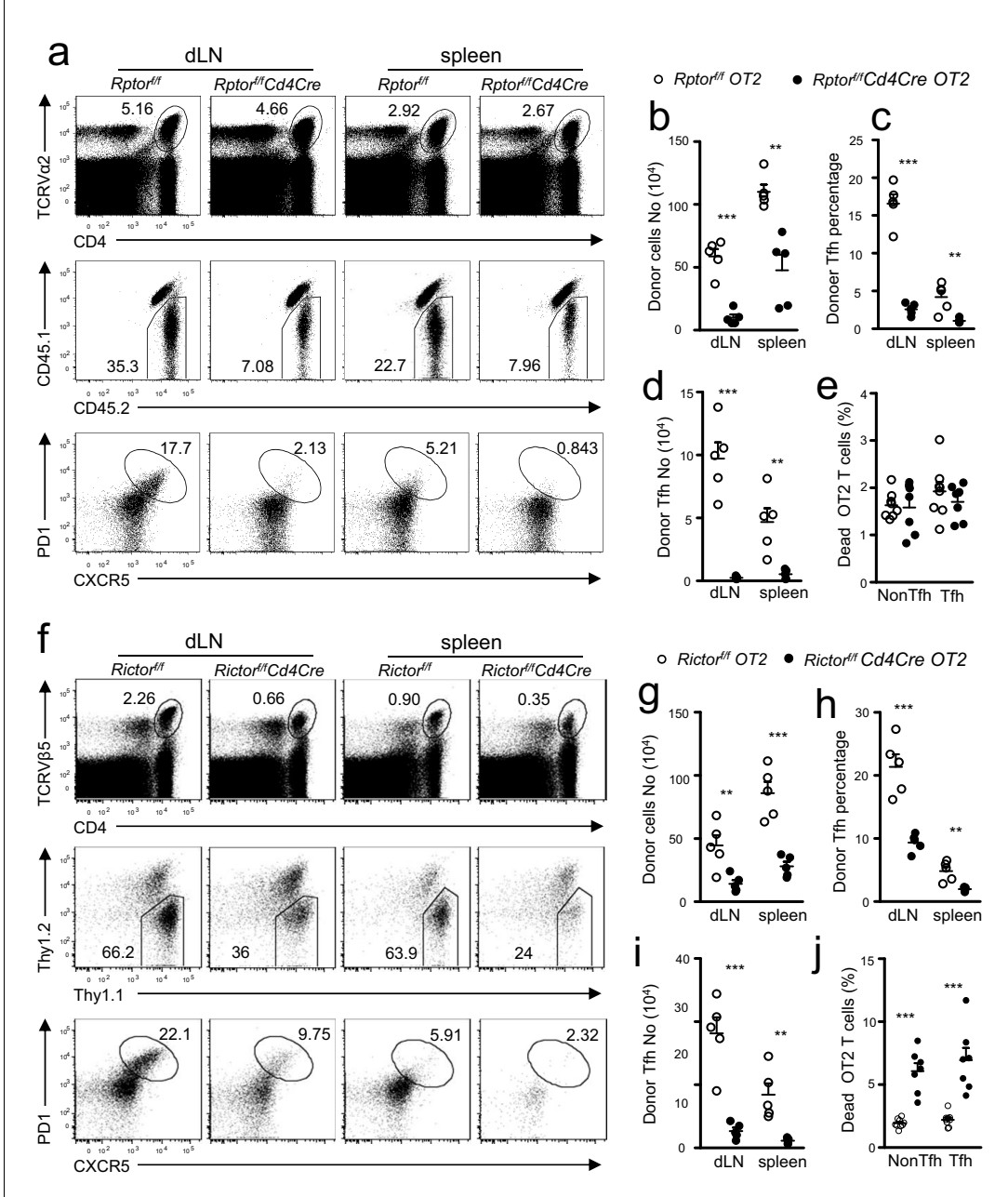

**Figure 6.** T cell intrinsic control of Tfh differentiation by mTORC1 and mTORC2. We injected CD45.1⁺CD45.2⁺ congenic mice *iv* with 1.5 × 10⁶ CD45.2⁺CD4⁺TCRVα2⁺ naïve OT2 T cells from *Rptor^f/f^-Cd4Cre-OT2* (a–e) or *Rictor^f/f^-Cd4Cre-OT2* (f–j) and control mice on day -1 and immunized them with OVA₃₂₃₋₃₃₉ peptide in CFA on day 0. We harvested dLNs and spleens on day 7 post immunization. (a, f) Representative dot plots of dLN cells and splenocytes. Top panels: CD4 and TCRVα2 staining. Middle panels: donor-derived CD45.1⁻CD45.2⁺ OT2 cells in the gated CD4⁺TCRVα2⁺ population. Bottom panels: donor-derived Tfh cells in gated donor OT2 cells. (b, g) Scatter plot shows absolute number of donor-derived *Rptor^f/f^-Cd4Cre* OT2 cells (b) and *Rictor^f/f^-Cd4Cre* OT2 cells (g). (c, h) Scatter plot shows Tfh percentages within donor-derived *Rptor^f/f^-Cd4Cre* OT2 cells (c) and *Rictor^f/f^-Cd4Cre* OT2 cells (h). (d, i) Scatter plot shows total Tfh cell numbers derived from *Rptor^f/f^-Cd4Cre* OT2 (d) and *Rictor^f/f^-Cd4Cre* OT2 cells (i). (e, j) Scatter plot shows death rates of donor-derived OT2 T cells. Cell death was determined by staining with Live/Dead fixable violet dead cell stain. Data shown represent or are calculated from two experiments. **p<0.01; ***p<0.001 determined by two-tailed unpaired Student *t*-test.

The following figure supplement is available for figure 6:

**Figure supplement 1.** Phenotypic analysis of *Rptor^f/f^*- or *Rictor^f/f^-Cd4Cre* OT2 T cells.

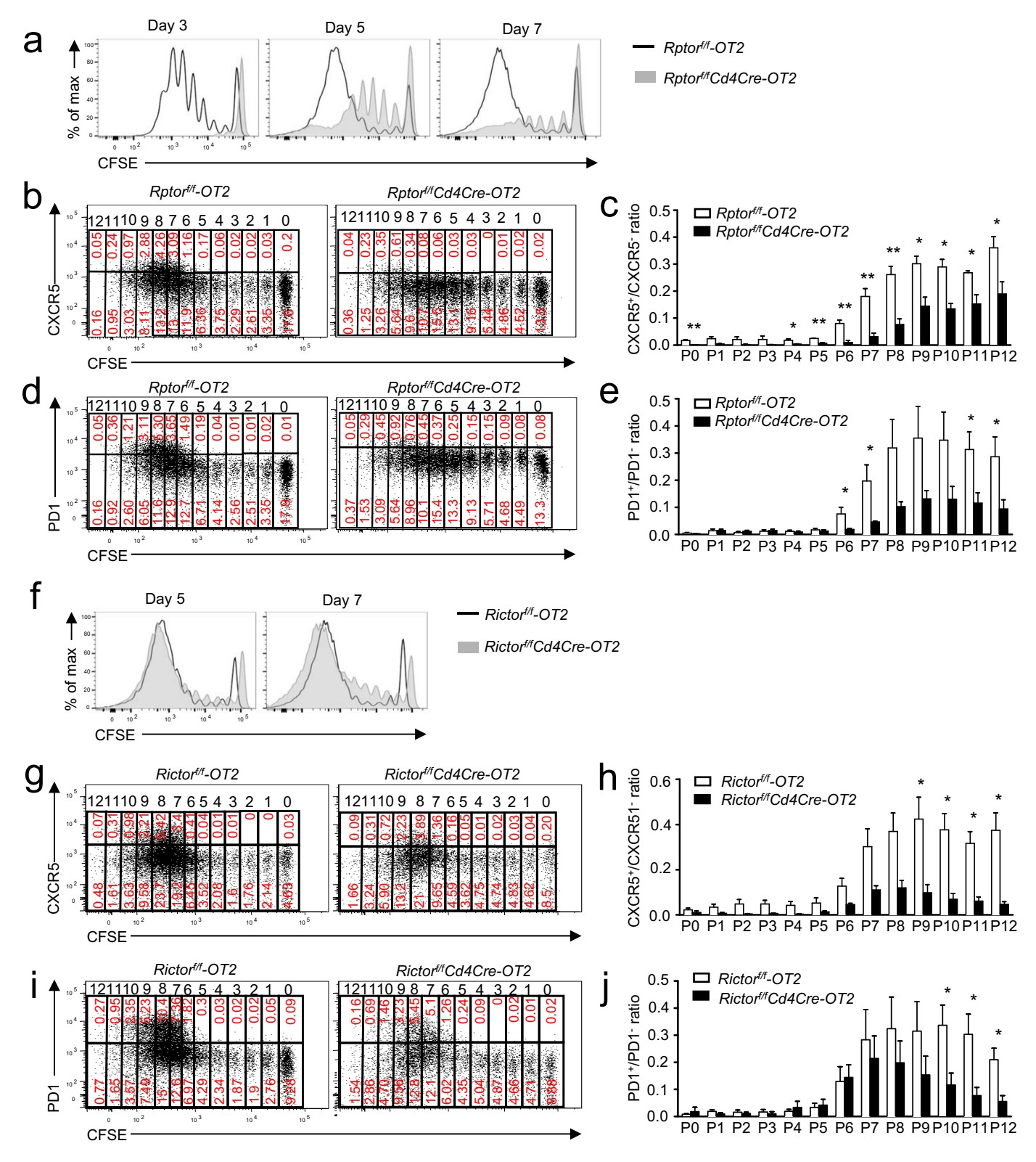

**Figure 7.** Effects of mTORC1 and mTORC2 deficiency on proliferation-associated Tfh differentiation. We injected CD45.1⁺CD45.2⁺ congenic mice *iv* with 1.5 × 10⁶ CD45.2⁺Vα2⁺CD4⁺ WT, *Rptor^{f/f}-Cd4Cre* (a–e), or *Rictor^{f/f}-Cd4Cre* (f–j) naïve OT2 T cells on day -1 and immunized them with OVA₃₂₃₋₃₃₉ peptide in CFA on day 0, harvesting dLNs on indicated days after immunization. (**a**) Overlaid histograms showing CFSE intensity in CD45.1⁻CD45.2⁺CD4⁺TCRVα2⁺ donor-derived WT and *Rptor*-deficient OT2 T cells 3, 5, and 7 days after immunization. (**b, d**) Representative dot plots

*Figure 7 continued on next page*

*Figure 7 continued*

of CXCR5 (**b**) and PD1 (**d**) staining and CFSE intensity in donor-derived WT and *Rptor*-deficient OT2 cells from dLNs on day 7 after immunization. (**c, e**) Bar graphs represent mean ± SEM of CXCR5$^+$/CXCR5$^-$ ratios (**c**, WT, n = 3; KO, n = 3) and PD1$^+$/PD1$^-$ ratios (**e**, WT, n = 4; KO, n = 4) in individual cell divisions of donor-derived WT and *Rptor*-deficient OT2 T cells. (**f**) Overlaid histograms showing CFSE intensity in CD45.1$^-$CD45.2$^+$CD4$^+$TCRVα2$^+$ donor-derived WT and *Rictor*-deficient OT2 T cells 5 and 7 days after immunization. (**g, i**) Representative dot plots of CXCR5 (**g**) and PD1 (**i**) staining and CFSE intensity in donor-derived WT and *Rictor*-deficient OT2 cells from dLNs on day 7 after immunization. (**h, j**) Bar graphs represent mean ± SEM of CXCR5$^+$/CXCR5$^-$ ratios (**h**, WT, n = 4; KO, n = 3) and PD1$^+$/PD1$^-$ ratios (**j**, WT, n = 5; KO, n = 4) in individual cell divisions of donor-derived WT and *Rictor*-deficient OT2 T cells. Data shown represent or are calculated from three independent experiments. *p<0.05; **p<0.01 (Student's *t* test).

The following figure supplement is available for figure 7:

**Figure supplement 1.** Effects of mTORC1 and mTORC2 deficiency on antigen-induced T cell activation in vitro.

However, the PD1$^+$/PD1$^-$ and CXCR5$^+$/CXCR5$^-$ ratios in individual divisions were much lower for Raptor-deficient OT2 T cells than for WT controls. Thus, Raptor/mTORC1 not only promoted proliferation of CD4 T cells to reach the needed divisions after activation to become Tfh cells but also played an important role for expanded T cells to differentiate into Tfh cells.

Unlike Raptor/mTORC1 deficiency, Rictor-deficient OT2 T cells proliferated and upregulated CD69 and CD25 at similar rates/levels compared with controls following in vitro TCR stimulation (*Figure 7—figure supplement 1c,d*). CD44 levels could also be upregulated, though they remained lower than stimulated WT controls. Furthermore, after immunization, proliferation of adoptively transferred Rictor-deficient OT2 T cells was not obviously impaired compared to WT controls in recipient mice (*Figure 7f*). However, their CXCR5$^+$/CXCR5$^-$ ratios in individual divisions after six divisions were decreased compared with WT controls (*Figure 7g,h*). The PD1$^+$/PD1$^-$ ratios in Rictor-deficient OT2 T cells were not different from WT control in division 6 but displayed a tendency of decline after division 7, although the decreases were only statistically significant after division 10 (*Figure 7i,j*). Together, these observations suggested that Tfh differentiation, rather than T cell activation and proliferation, selectively required Rictor/mTORC2.

## mTORC2 promotes Tfh differentiation via activating Akt and upregulating TCF1

Because mTORC1-deficient T cells were defective in proliferation following T cell activation, we focused on investigating mechanisms by which mTORC2 promoted Tfh differentiation. Recent studies have revealed that the PI3K/Akt pathway plays an important role in Tfh differentiation (*Gigoux et al., 2009*; *Stone et al., 2015*; *Rolf et al., 2010*). Following TCR engagement, we observed apparently decreased Akt phosphorylation at Ser473, an mTORC2-dependent event, but not S6 phosphorylation, an mTORC1-dependent event, in Rictor/mTORC2-deficient T cells (*Figure 8a, b*). Concordant with decreased Akt phosphorylation, GSK-3β phosphorylation at Ser9 residue, an Akt mediated event, also decreased in these cells (*Figure 8a*). Thus, mTORC2 deficiency resulted in impaired Akt activation in CD4 T cells following TCR stimulation.

To examine whether decreased Akt S473 phosphorylation contributed to defective Tfh differentiation, we adoptively transferred WT or *Rictor*-deficient OT2 T cells (CD45.2$^+$) transduced with retroviruses coexpressing either the Akt S473D phosphomimetic mutant and GFP or GFP alone (Migr1) into WT C57BL6/J recipient mice, followed by immunization with OVA$_{323-339}$ peptide in CFA. Seven days after immunization, we observed a two-fold increase of PD1$^+$CXCR5$^+$ Tfh cells within GFP$^+$-TCRVα2$^+$Vβ5$^+$CD4$^+$*Rictor*-deficient OT2 T cells expressing AktS473D compared with those expressing GFP alone (*Figures 8c,d*). Such an effect of AktS473D on Tfh differentiation was correlated with its ability to upregulate Bcl-6 expression (*Figure 8e*) and to improve cell survival (*Figure 8—figure supplement 1*). Of note, *Rictor*-deficient OT2 T cells expressing AktS473D were still about 50% fewer Tfh cells than WT OT2 T cells (*Figure 8c,d*). Together, these observations suggested that mTORC2 promoted Tfh differentiation via both Akt-dependent and -independent mechanisms.

Recent studies reveal that the Wnt/β-catenin/TCF1 axis plays an important role in Tfh differentiation, at least by increasing Bcl-6 and ASCL2 expression (*Choi et al., 2015*; *Liu et al., 2014*; *Xu et al., 2015*). Because GSK3β negatively controls the Wnt/β-catenin/TCF1 axis by directly phosphorylating β-catenin to trigger its degradation, and Akt phosphorylates and inactivates GSK3β

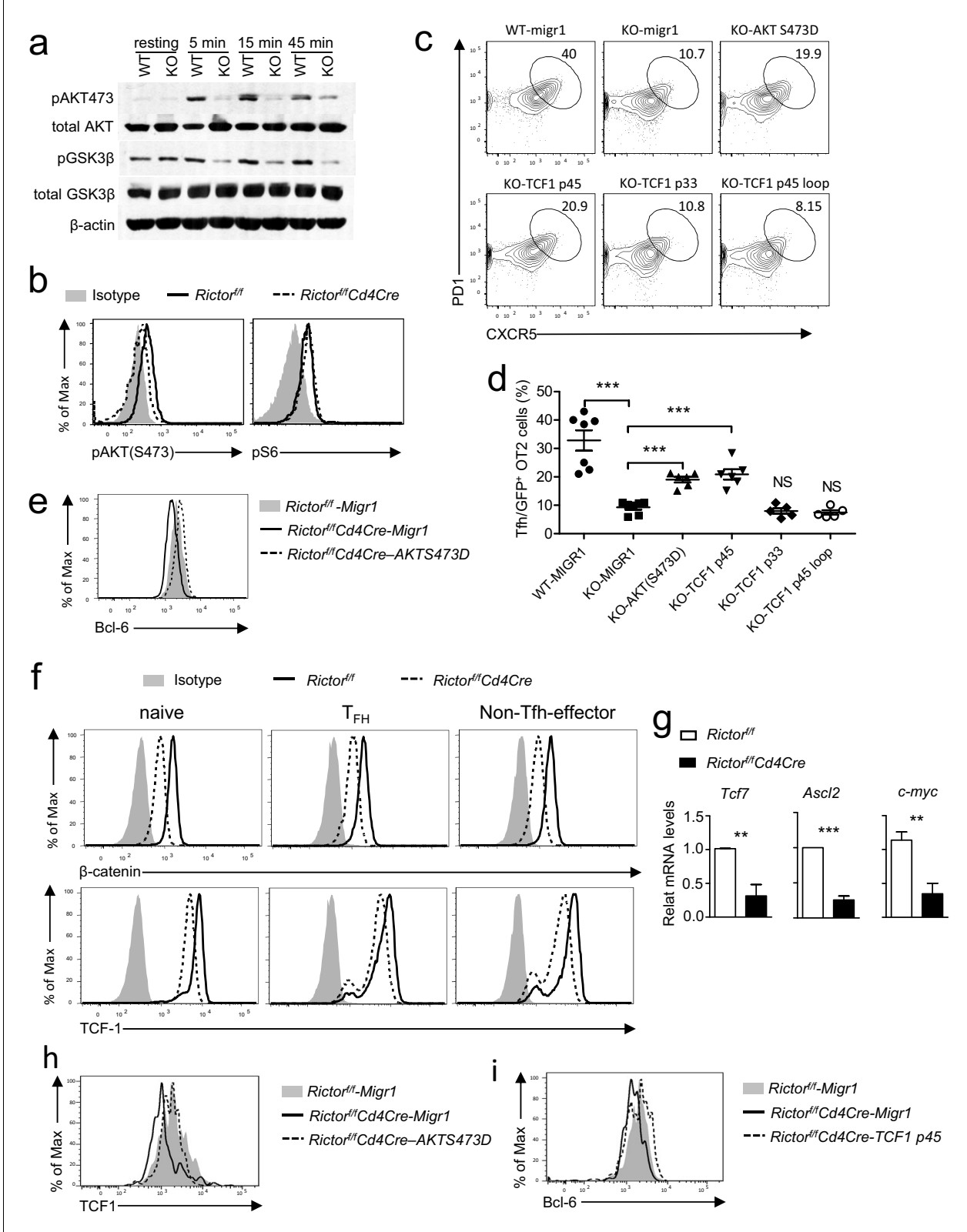

**Figure 8.** mTORC2 controls AKT and TCF1 to promote Tfh differentiation. (a) We rested WT and *Rictor^f/f-Cd4Cre* CD4 T cells in PBS for 30 min, then simulated them with an anti-CD3ε antibody (500 A2) for the indicated times. We subjected cell lysates to immunoblotting analysis with the indicated antibodies. (b) We stimulated WT and *Rictor^f/f-Cd4Cre* naive CD4⁺ T cells with plate-bound anti-CD3 and anti-CD28 for 24 hr. We used intracellular staining and FACS analysis to determine Akt S473 and S6 phosphorylatio. (c–e). WT or *Rictor^f/f-Cd4Cre* OT2 T cells were transduced with retrovirus
*Figure 8 continued on next page*

*Figure 8 continued*

expressing AKT S473D, TCF1p45, TCF1p33, and TCF1p45Loop34 plus GFP or GFP alone (Migr1) 48 hr after stimulating them with anti-CD3 and anti-CD28. Twenty-four hours after transduction, we intravenously injected cells injected into congenic C57B6/J mice immunized with OVA$_{323-339}$ peptide in CFA about 18 hr after transfer and analyzed them 7 days after immunization. (c) Representative contour plots showing CXCR5 and PD1 expression in live-gated GFP$^+$CD4$^+$CD8$^-$TCRVα2$^+$Vβ5$^+$B220$^-$CD11b$^-$Gr1$^-$ donor-derived OT2 T cells. (d) Scatter plots representing mean ± SEM of CXCR5$^+$PD1$^+$ cells within GFP$^+$ donor OT2 cells. (e) Overlaid histograms showing Bcl-6 levels in GFP$^+$OT2 T cells 48 hr after transduction. (f) Overlaid histograms showing intracellular β-catenin and TCF1 staining in gated Tfh (CD4$^+$CXCR5$^+$PD1$^+$), naïve (CD4$^+$CD44$^-$), and non-Tfh effector (CD4$^+$CD44$^+$CXCR5$^-$PD1$^-$) from mLNs in WT and *Rictor$^{f/f}$*-*Cd4Cre* mice. (g) Relative mRNA levels of indicated molecules in Tfh cells sorted from PPs and mLNs from WT and *Rictor$^{f/f}$*-*Cd4Cre* mice. (h) Overlaid histograms showing TCF1 levels in GFP$^+$OT2 T cells 48 hr after transduction with retrovirus expressing AKT S473D or GFP. (i) Overlaid histograms showing Bcl6 levels in GFP$^+$OT2 T cells 48 hr after transduction with retrovirus expressing TCF1 p45 or GFP. Data shown are representative of three (a–f, h, i) or calculated from five (g) independent experiments. **p<0.01; ***p<0.001 (Student's *t* test).

The following figure supplement is available for figure 8:

**Figure supplement 1.** Expression of Akt S473D improved survival of *Rictor* deficient CD4 T cells.

---

(*Staal et al., 2008*; *Hur and Zhou, 2010*), we examined β-catenin and TCF1 expression in *Rictor*-deficient CD4 T cells. As *Figure 8f* shows, *Rictor*-deficient Tfh, naïve, and non-Tfh effector CD4 T cells expressed lower levels of β-catenin and TCF1 than did their WT controls. Additionally, both mRNA levels of TCF1 (*Tcf7*) and β-catenin responsive genes *ASCL2* and *c-Myc* also decreased in *Rictor*-deficient Tfh cells (*Figure 8g*), indicating that mTORC2 deficiency diminished β-catenin and TCF1 expression in CD4 T cells. Importantly, expression of AktS473D increased TCF1 expression (*Figure 8h*) and overexpression of the TCF1-p45 long isoform capable of binding β-catenin, but not the TCF1-p33 short isoform lacking β-catenin-interacting domain, increased Tfh differentiation about two-fold (*Figure 8c,d*) . TCF1 has recently demonstrated intrinsic HDAC activity embedded within a 30 amino acid (Q192-L221) region (*Xing et al., 2016*). Forced expression of a TCF1-Loop34 mutant, which lacks the HDAC domain, failed to show a 'rescue' effect, suggesting a contribution from the TCF1 HDAC activity for Tfh differentiation (*Figure 8c,d*). Correlated with their ability to rescue Tfh differentiation, TCF1-p45 was able to upregulate Bcl-6 expression in *Rictor* deficient OT2 T cells (*Figure 8i*). Together, these observations suggest that decreased TCF1 expression may at least partly contribute to the failure of Rictor/mTORC2-deficient CD4 T cells to differentiate to Tfh cells.

## Discussion

Understanding regulation of Tfh cell differentiation is crucial for developing new strategies to elicit protective immunity effectively and to improve treatment of autoimmune diseases. Although mTOR has proven important for Th1, Th2, and Th17 differentiation, its role in Tfh differentiation has been controversial. A recent study showed that inhibition of mTORC1 with rapamycin or reduction of mTOR expression with shRNA promotes Tfh differentiation following lymphocytic choriomeningitis virus infection, concluding that mTORC1 negatively controls Tfh differentiation (*Ray et al., 2015*). However, another study has found that a hypomorphic mutation of mTOR reduces Tfh and GC responses in mice immunized with sheep red blood cells, concluding that mTOR promoted Tfh differentiation (*Ramiscal et al., 2015*). Additionally, rapamycin has also been found to inhibit GC formation and Ig class-switch in B cells upon influenza infection, although whether rapamycin directly acts on T cells, B cells, or both is unclear (*Keating et al., 2013*). While these two studies suggest a positive role for mTOR in Tfh differentiation, they neither firmly establish that mTOR intrinsically controls CD4 T cell differentiation in Tfh cells nor determine the roles of mTORC1 and mTORC2 in Tfh differentiation.

In this report, we demonstrate that T cell-specific ablation of mTOR severely decreases constitutive Tfh and GC B cell formation in mLNs and PPs and reduced serum IgG and fecal IgA levels. Moreover, mice with T cell-specific mTOR deficiency fail to mount effective Tfh and GC responses following immunization with a T cell-dependent antigen. Our study provides genetic evidence that mTOR plays an intrinsically crucial role in CD4 T cells for their differentiation to Tfh cells and for promotion of GC responses. Our data also reveal that both mTORC1 and mTORC2 are important for Tfh differentiation. We have shown that ablation of either Raptor/mTORC1 or Rictor/mTORC2 in T

cells reduces Tfh differentiation and GC B cell formation in mLNs and PPs under the steady state as well as in the spleen following immunization with NP–CGG. Although both mTORC1 and mTORC2 contribute to Tfh differentiation, mTORC1 deficiency appears to exert greater impact on Tfh differentiation than mTORC2 deficiency in antigen-induced responses, as NP-specific antibody responses are more severely affected in mTORC1-deficient mice than in mTORC2-deficient mice.

Using adoptive transfer of OT2 CD4 T cells, we have shown that, following antigenic stimulation, differentiation of naïve OT2 CD4 T cells to Tfh cells occurs after six divisions. Most mTORC1 deficient CD4 T cells fail to proliferate, suggesting that an important function of mTORC1 is to promote CD4 T cell proliferation, which is consistent with its role in causing T cells to enter into the cell cycle (*Yang et al., 2013*). In addition, beyond its role in cell proliferation, mTORC1 likely promotes Tfh differentiation, because some mTORC1-deficient CD4 T cells can reach the required cell divisions but remain defective for turning into Tfh cells. mTORC1 has been shown to be able to upregulate ICOS expression in CD4 T cells to prevent anergy (*Xie et al., 2012*). Given the importance of ICOS expression in Tfh differentiation (*Kang et al., 2013*), mTORC1 could operate as a positive feedback mechanism to promote Tfh differentiation by upregulating ICOS expression.

Our data provide the first evidence that mTORC2 also plays important roles for Tfh differentiation and GC-formation. Unlike mTORC1 deficiency, mTORC2 deficiency does not obviously affect T cell proliferation in vitro and in vivo, suggesting that impaired proliferation might not cause defective Tfh differentiation of mTORC2 CD4 T cells. Different from mTORC1, mTORC2 promotes T cell survival during immune responses. Our data suggest that mTORC2 may promote Tfh differentiation at least by increasing Akt activity. mTORC2-deficient CD4 T cells contain decreased Akt phosphorylation at S473, accompanying reduced enzymatic activity reflected by decreased GSK3β phosphorylation at Ser9 and increased CD4 T cell death. Importantly, reconstitution of Akt S473 phosphomimetic mutant AktS473D into mTORC2-deficient CD4 T cells can improve their survival, restore Bcl-6 expression in these cells, and partially reverse their defect in Tfh differentiation. Akt itself phosphorylates numerous substrates to control their activities and subcellular localization. Among them, Foxo1 is known to inhibit Tfh differentiation by directly suppressing Bcl-6 transcription (*Stone et al., 2015*). Because Akt phosphorylates Foxo1, leading to its sequestration in the cytosol, decreased Akt activity could relieve Bcl-6 from Foxo1-mediated suppression. Thus, mTORC2 may increase Akt activity to relieve Foxo1-mediated repression of Bcl-6 expression during Tfh differentiation. In addition to Akt, mTORC2 phosphorylates several other substrates such as PKCα, PKCθ, and SGK1. Whether these substrates contribute to mTORC2-mediated Tfh differentiation remains a question for future investigation.

Our data suggested another potential Akt-mediated mechanism is the regulation of the Wnt/β-catenin/TCF1 axis. Recent studies have demonstrated that TCF1 plays critical roles in Tfh differentiation by directly controlling expression of Tfh-promoting genes such as Bcl-6 and ASCL2 (*Wu et al., 2015*; *Xu et al., 2015*; *Choi et al., 2015*). GSK3β negatively controls Wnt/β-catenin signaling by directly phosphorylating β-catenin at multiple sites, leading to the degradation of β-catenin (*Staal et al., 2008*). Akt-mediated phosphorylation negatively controls GSK3β activity at Ser9 (32). Decreased β-catenin levels and TCF1 expression suggest that the Wnt/β-catenin/TCF1 axis is impaired in mTORC2-deficient CD4 T cells, which could be because decreased GSK3β phosphorylation in mTORC2-deficient CD4 T cells causes increased GSK3β activity. Reconstituting these cells with full-length TCF1 partially restores their ability to differentiate into Tfh cells, suggesting that mTORC2 promotes TCF1 expression to enhance Tfh differentiation. It was noteworthy that the TCF1 p33 isoform failed to rectify Rictor deficiency-caused Tfh defects, highlighting a requirement for TCF1–β-catenin interaction.

Our data are consistent with previous studies that have demonstrated important roles in Tfh differentiation and GC-responses for PI3K signaling, which is involved in the induction of key molecules such as Bcl-6, IL-4 and IL-21 *Gigoux et al., 2009*; *Rolf et al., 2010*; *Bauquet et al., 2009*; *Choi et al., 2011*. Upregulation of PI3K signaling by miR-17–92 promotes Tfh cell differentiation (*Kang et al., 2013*; *Baumjohann et al., 2013*), while PTEN's negative control of PI3K signal or Foxp1's downregulation of ICOS inhibits Tfh differentiation (*Wang et al., 2014*; *Rolf et al., 2010*). Because both mTORC1 and mTORC2 are activated downstream of PI3K, our data suggest that mTORC1/mTORC2 are critical downstream effectors of the PI3K/Akt pathway for Tfh differentiation. Further studies should illustrate how different receptors and intracellular signals dynamically activate and regulate mTOR during Tfh differentiation and how altered mTOR signaling may either

contribute to pathogenesis of autoimmune diseases caused by uncontrolled Tfh and GC responses or improve efficacy during vaccination.

# Materials and methods

## Mice

We purchased $Mtor^{f/f}$(73), $Rptor^{f/f}$(74), $Rictor^{f/f}$ (*Magee et al., 2012*), $Rag2^{-/-}$, and CD4-Cre (*Lee et al., 2001*) mice from the Jackson Laboratory and Taconic Farms. $Rosa26^{CreER}$ mice were previously described (*Shapiro-Shelef et al., 2005*). We used 6- to 12-week old $Mtor^{f/f}$-*Cd4Cre*, $Rptor^{f/f}$-*Cd4Cre*, $Rictor^{f/f}$-*Cd4Cre*, and $Rptor^{f/f}$-$Rosa26^{CreER}$ mice and their respective Cre-negative (WT) littermates for experiments. We performed all experiments according to protocols approved by the Duke University Institute Animal Care and Use Committee.

## Reagents, plasmids, and antibodies

We supplemented Iscove's modified Dulbecco's medium (IMDM) with 10% (vol/vol) FBS, penicillin/streptomycin, and 50 µM 2-mercaptoethanol (IMDM-10). We determined cell death using Live/Dead Fixable Violet Dead Cell Stain (Invitrogen, Carlsbad, CA) according to the manufacturer's protocol and analyzed cell death flow cytometry. We purchased fluorescence-conjugated anti-mouse CD4 (GK1.5), TCRβ (H57-597), TCRVα2 (B20.1), TCRVβ5 (MR9-4), CD3 (145–2 c11), CD25 (PC61), CD44 (IM7), CD62L (MEL-14), CD45.1 (A20), CD45.2 (104), Thy1.1 (OX-7), Thy1.2 (58-2.1), CD69 (H1.2F3), CD122 (TM-β1), CD127 (SB/199), B220 (RA3-6B2), CXCR5 (L138D7), ICOS (C398.4A), PD-1 (RMP1-30), and GL7 (GL7) antibodies from BioLegend (San Diego, CA). We purchased anti-mouse FAS (Jo2) antibodies from BD Biosciences (San Jose, CA) and anti-mouse CD93 (AA4.1), Foxp3 (FJK-16s), and Bcl-6 (BCL-DWN) antibodies from eBioscience (San Diego, CA ). We purchased anti-phosphor AKT S473 (D9E), phosphor S6 (D57.2.2E), β-catenin (6B3), and TCF1 (C63D9) antibodies from Cell Signaling Technology (Danvers, MA). AKT S473D and TCF1 p45, TCF1 p33, and TCF1-Loop34 mutant retroviral vectors have been reported previously (*O'Brien et al., 2011*; *Xing et al., 2016*).

## Flow cytometry

We used standard protocols to prepare single-cell suspensions from the spleen, mLNs, and PPs of mice. After lysis of red blood cells with the ACK buffer, we resuspended cells in IMDM-10 and then stained them with antibodies in PBS containing 2% FBS. We performed intracellular staining for Foxp3, Bcl-6, β-catenin, and TCF1 using the eBioscience Foxp3 Staining Buffer Set and intracellular staining for AKT S473 and S6 using the BD Biosciences Cytofix/Cytoperm and Perm/Wash solutions. We collected all flow cytometry data using a FACS Canto-II (BD Biosciences) and analyzed them using FlowJo.

## Total and subtype Ig quantification

We collected and weighed fresh fecal pellets from mice and resuspended them in PBS at 100 mg/ml. After vortexing the pellets for 5 min and centrifuging them at 3000 g for 10 min, we transferred the supernatant to new tubes containing a protease inhibitor cocktail and stored them at −80°C until use. We used ELISA to measure the total IgA levels in fecal preparations and serum IgM, total IgG, IgG1, IgG2b, IgG2c, and IgG3 levels. In brief, we added100µl of appropriately diluted fecal or serum samples to 96-well plates precoated with anti-mouse Igκ and Igλ antibodies (2 µg/ml; SouthernBiotech, Birmingham, AL) in 0.1 M carbonate buffer (pH 9.0) at 4°C overnight. We determined total and subtype Ig concentrations using HRP-conjugated goat anti-mouse total or Ig subtype antibodies (SouthernBiotech). We computed Ig relative levels by the OD450 values.

## Immunization, antibody responses, and GC detection

We immunized the mice with a single i.p. injection of 20 µg of 4-hydroxy-3-nitrophenylacetyl conjugated chicken gamma globulin (NP$_{17}$-CGG) in alum as previously described (*Ci et al., 2015*). We collected serum on day 7, 14, and 21 postimmunization. To measure NIP-specific IgM and IgG levels, we appropriately diluted the serum and added it into ELISA plates (Corning, New York, NY) precoated with 50 µl 2 µg/ml NIP$_7$-BSA or NIP$_{26}$-BSA in 0.1 M carbonate buffer (pH 9.0) at 4°C

overnight. After incubation and multiple washes, we used HRP-conjugated goat anti-mouse IgM and IgG to detect NIP-specific IgM and IgG, respectively.

To visualize GC, we fixed spleens from mice 14 days after immunization with 4% PFA for 24 hr and incubated them in 30% sucrose solution for another 24 hr. After freezing the spleens in OCT embedding medium, we cut 10 μm sections and blocked them with PBS containing 3% BSA for 30 min before incubating them with biotinylated Peanut Agglutinin (PNA, Vector Laboratories, Burlingame, CA) at room temperature for 1 hr. We washed slides with PBS and stained them with Violet 421-anti-B220 (Biolegend, clone RA3-6B2, 2 μg/ml), FITC-anti-Thy1.2 (Biolegend, clone 30-H12, 5 μg/ml), and Alex 594-conjugated streptavidin (2 μg/ml) overnight at 4℃. Finally, we covered the stained sections with slow-fade diamond antifade mountant (Life Technologies, Carlsbad, CA). We collected fluorescence images using a Zeiss Axio imager wide-field fluorescence microscope with 5X and 20X objectives and used Photoshop (Adobe Systems, San Jose, CA) for postacquisition brightness and contrast processing. We used MetaMorph image analysis software to quantify germinal center (PNA$^+$Thy1.2$^-$) sizes.

## In vivo tfh and GC-B cell responses after immunization

We transferred 1.5 million CFSE labeled or unlabeled CD45.2$^+$ naïve WT, Rptor$^{f/f}$-Cd4Cre or Rictor$^{f/f}$-Cd4Cre OT2 cells, or retrovirally transduced OT2 cells into congenic CD45.1$^+$CD45.2$^+$ recipients, which we subsequently immunized with 100 μg/mouse OVA$_{323-339}$ peptide emulsified in CFA 18–24 hr later. We used flow cytometry to identify Tfh cells in draining recipient LNs and spleens at indicated times after immunization.

## Realtime RT-PCR

For RNA preparation, we immediately lysed sorted viable cells in Trizol and made cDNA using the iScript Select cDNA Synthesis Kit (Biorad). We conducted and analyzed real-time quantitative PCR as previously described (*Shin et al., 2012*). We normalized and calculated expressed levels of target mRNAs with *β-actin* using the $2^{-\Delta\Delta CT}$ method, using the following primers: *Tcf7* Forward: CCCCAGC TTTCTCCACTCTA, Reverse: GCTGCCTGAGGTCAGAGAAT; *Ascl2* Forward: CGCTGCCCAGAC TCATGCCC, Reverse: GCTTTACGCGGTTGCGCTCG; *c-myc* Forward: AGTGCTGCATGAGGAGA-CAC, Reverse: GGTTTGCCTCTTCTCCACAG; *β-actin* Forward: TGTCCACCTTCCAGCAGATGT, Reverse: AGCTCAGTAACAGTCCGCCTAGA.

## Western blot

We rested splenocytes in DPBS for 30 min and then stimulated or unstimulated them with anti-CD3 (500 A2) for different times, after which we immediately lysed them in lysis buffer (1% nondiet P-40, 150 mM NaCl, 50 mM Tris, pH 7.4) with freshly added protease and phosphorylases inhibitors. We subjected cell lysates were subjected to immunoblotting analysis using indicated antibodies.

## Retroviral transduction

We used the Phoenix-Eco packaging cell line to make retroviruses for Migr1, AKT S473D, TCF1 p45 and p33 isoforms, and TCF1-Loop34. Phoenix-Eco cells, kindly provided by Dr. Gary Nolan from Stanford University, were authenticated by STR profiling and were tested free of mycoplasma contamination. We stimulated three million splenocytes in 24-well plates in 1 ml IMDM-10 with anti-CD3 (1 μg/mL) and anti-CD28 (500 ng/mL) for 40 hr. After replacing 500 μl cultural medium with retroviral supernatants containing polybrene (5 μg/mL final concentration), we spin-infected cells at 22℃, 1250 g, for 1.5 hr. After incubating culture supernatants at 37℃ for 6 hr, we replaced them with fresh IMDM-10 and cultured cells for an additional 48 hr before use.

## Statistical analysis

We present data as mean ± SEM; we determined statistical significance using the two-tailed Student $t$ test. We define $p$ values as follows: *$p < 0.05$; **$p < 0.01$; ***$p < 0.001$.

## Acknowledgements

We thank the Flow Cytometry Facility at Duke Cancer Institute for sorting services. Our work is supported by grants from NIAID, NIH (R01AI079088 and R01AI101206 for XPZ and AI112579, AI115149, and AI119160 for HHX). All authors declare no conflict of interest.

## Additional information

### Funding

| Funder | Grant reference number | Author |
| --- | --- | --- |
| National Institutes of Health | R01AI079088 | Xiao-Ping Zhong |
| National Institutes of Health | R01AI101206 | Xiao-Ping Zhong |
| National Institutes of Health | R01AI112579 | Hai-Hui Xue |
| National Institutes of Health | R01AI115149 | Hai-Hui Xue |
| National Institutes of Health | R01AI119160 | Hai-Hui Xue |

The funders had no role in study design, data collection and interpretation, or the decision to submit the work for publication.

### Author contributions

JY, Conception and design, Acquisition of data, Analysis and interpretation of data, Drafting or revising the article; XL, YP, PC, Acquisition of data, Analysis and interpretation of data; JW, HH, JG, Designed and performed experiments, and analyzed data; H-HX, Contributed unpublished critical reagents; X-PZ, Conception and design, Analysis and interpretation of data, Drafting or revising the article

### Author ORCIDs

Xiao-Ping Zhong, http://orcid.org/0000-0002-4619-8783

### Ethics

Animal experimentation: This study was performed in strict accordance with the recommendations in the Guide for the Care and Use of Laboratory Animals of the National Institutes of Health. All of the animals were handled according to approved institutional animal care and use committee (IACUC) protocols A051-16-03 and A095-13-04) of Duke University.

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
