## [Decision Letter]

Thank you for submitting your article "Critical roles of mTOR Complex 1 and 2 for T Follicular Helper Cell Differentiation and Germinal Center Responses" for consideration by *eLife*. Your article has been favorably evaluated by Tadatsugu Taniguchi as the Senior Editor and three reviewers, one of whom is a member of our Board of Reviewing Editors.

The reviewers have discussed the reviews with one another and the Reviewing Editor has drafted this decision to help you prepare a revised submission. While they find your work of potential interest, they have raised substantial issues that need to be addressed before we can consider publication in *eLife*. Should further experimental data allow you to address these issues substantively, we would be happy to look at a revised manuscript. Please note our editorial policy that if the revision will likely take more than two months, the revised version will be considered as a new submission. Therefore, we request that you respond with details of how you propose to deal with the criticisms and a timetable for the additional work necessary to produce an acceptable revised submission.

Summary:

The manuscript shows an interesting set of data that identify roles for mTor, and particularly for the mTorc1- and mTorc2-related proteins Raptor and Rictor, in Tfh differentiation, with some insights into the possible downstream mechanisms contributing to the role of Rictor in Tfh differentiation. While the data are incremental in that they take forward the directions indicated by recent literature, they do provide important new information well worth publication.

Essential revisions:

However, it would be useful to have three sets of concerns addressed substantively.

1) The first concern relates to the fact that the roles of actual mTor, Raptor or Rictor functions during Tfh differentiation are not addressed as distinct from prior alteration in naïve T cell programs due to these deficiencies in the T cell lineage, despite some degree of evidence for the latter. Data with deletion/inhibition of mTOR, Raptor and Rictor in CD4 T cells just prior to their functional assays would be useful.

2) Antibody responses, a major functional readout, would be best measured and expressed as concentrations rather than as absorbance values. Further, it may be noted that the B6 genotype lacks IgG2a and instead expresses IgG2c (Journal of Immunological Methods (1998) 212:187-192).

3) Since more regulatory T cells (Tregs) have been reported in CD4 T cells lacking mTor, it is plausible that reductions in Tfh cells/functions could be related to effects on Tregs. Data exploring this possibility would be useful.

4) Another concern is about the fact that, while the quantitative differences in germinal center outcomes due to mTor-null, Raptor-null, or Rictor-null CD4 T cells are noted and addressed to some extent, the qualitative differences in terms of the radically different effects on specific Ig isotypes are not addressed at all by way of either examining if Raptor/Rictor truly function in the mTor pathway for these effects, or if Raptor and Rictor deficiencies together phenocopy mTor deficiency, or whether the differentiation and/or the effector receptor/cytokine profiles of residual Tfh cells differ between mTor-null, Raptor-null and Rictor-null genotypes. Characterization of residual Tfh cells in the various genotypes in terms of RNA-Seq and/or differentiation markers and cytokine and effector molecule estimations, and the phenotypic characterization of mTor+Raptor/Rictor and Raptor/Rictor genotypes would be useful.

5) Since the data indicate a role for mTor/Raptor in regulating Tfh proliferation and survival, careful estimation of Tfh numbers over the entire period of the immune response and quantitative comparison of cell death and survival rates would be useful.

6) The final concern relates to the mechanisms by which Rictor mediates Tfh differentiation. Additional data would greatly help clarify these. Thus, to clarify the role of Akt/GSK3b/Wnt/b-catenin/TCF1 pathway, does AktS473D over-expression in Rictor-deficient OT2 cells rescue Tcf7 and Ascl2 expression? To clarify the role of the Tcf1 p45 loop, does Tcf1 p45 loop over-expression in Rictor-deficient OT2 cells induce Bcl6 expression? The finding that forced expression of a constitutively active Akt (AktS473D) restores Bcl6 expression may indicate (as the authors speculate) that mTorc2 regulates Tfh differentiation in an Akt-independent mechanism, it is also possible that the β-catenin/Tcf1 pathway contributes to Tfh differentiation independent of the Akt-Bcl6 axis. Experiments examining whether AktS473D expression in Rictor-deficient T cells rescues the expression of β-catenin as well as Tcf1, and whether over-expression of Tcf1 lead to Bcl6 restoration would be useful to address this issue.

---

## [Author Response]

*Essential revisions:*

*1) The first concern relates to the fact that the roles of actual mTor, Raptor or Rictor functions during Tfh differentiation are not addressed as distinct from prior alteration in naïve T cell programs due to these deficiencies in the T cell lineage, despite some degree of evidence for the latter. Data with deletion/inhibition of mTOR, Raptor and Rictor in CD4 T cells just prior to their functional assays would be useful.*

We appreciate the reviewer for raising this issue. We adoptively transferred T cells from WT and Rptorf/f-ERCre mice into Rag2 deficient mice. Recipients were injected with tamoxifen seven, eight, and eleven days and examined on day 14 after reconstitution. We have found that Rptorf/f-ERCre CD4 T cells in the mesenteric lymph nodes in tamoxifen injected mice contained less CXCR5^+^PD1^+^ Tfh cells than WT control cells. These data support that Raptor/mTORC1 plays an important role in Tfh differentiation. These data are shown in Figure 2—figure supplement 1.

In addition, we also generated lethal irradiation chimeric mice reconstituted with a mixture of WT and Rptorf/f-ERCre bone marrow cells. Six to eight weeks after reconstitution, recipient mice were treated with tamoxifen on days 1, 2, and 5 followed by euthanization of mice on day 8. We found that Rptorf/f-ERCre derived CD4 T cells in the mesenteric lymph nodes contained less ICOS^+^PD1^+^ cells than WT control cells. Because these experiments were performed a while ago, we regret that we did not include additional markers to analyze Tfh in more detail. For this reason, we do not plan to present the data in this manuscript.

We currently do not carry mTORf/f and Rictorf/f-ERCre or OX40Cre mice that would allow us to delete these molecules in mature T cells during activation. We hope that the reviewers and editors would agree that both experiments with deletion of Raptor in mature T cells support that mTORC1 promotes Tfh differentiation.

*2) Antibody responses, a major functional readout, would be best measured and expressed as concentrations rather than as absorbance values. Further, it may be noted that the B6 genotype lacks IgG2a and instead expresses IgG2c (Journal of Immunological Methods (1998) 212:187-192).*

We greatly appreciate the reviewer for pointing out this issue. We have deleted the IgG2a data. We used absorbance values for antibody responses in considering that we were comparing the relative differences between WT and test mice. We regret that we lost some of the serum samples generated in these experiments and did not measure IgG2c concentration. While we acknowledge that absorbance values are not as good as concentrations, we hope that the reviewer would agree with us that for the purpose of comparing the effects of T cell specific deletion of mTOR deficiency on B cell responses the data in their current form is acceptable.

*3) Since more regulatory T cells (Tregs) have been reported in CD4 T cells lacking mTor, it is plausible that reductions in Tfh cells/functions could be related to effects on Tregs. Data exploring this possibility would be useful.*

We have examined Treg and Tfr numbers in mTORC1 and mTORC2 deficient mice as advised. The new data are shown in Figure 3.

*4) Another concern is about the fact that, while the quantitative differences in germinal center outcomes due to mTor-null, Raptor-null, or Rictor-null CD4 T cells are noted and addressed to some extent, the qualitative differences in terms of the radically different effects on specific Ig isotypes are not addressed at all by way of either examining if Raptor/Rictor truly function in the mTor pathway for these effects, or if Raptor and Rictor deficiencies together phenocopy mTor deficiency, or whether the differentiation and/or the effector receptor/cytokine profiles of residual Tfh cells differ between mTor-null, Raptor-null and Rictor-null genotypes. Characterization of residual Tfh cells in the various genotypes in terms of RNA-Seq and/or differentiation markers and cytokine and effector molecule estimations, and the phenotypic characterization of mTor+Raptor/Rictor and Raptor/Rictor genotypes would be useful.*

We thank the reviewer for raising this interesting issue. We agree that detection of mTOR function independent of Raptor/rictor or mTOR-independent function of Raptor/Rictor is interesting. To fully address these issues, we have to generate double knock out mice which requires complex time consuming breeding. We hope the editors and reviewers would agree that we can study these issues in the future.

*5) Since the data indicate a role for mTor/Raptor in regulating Tfh proliferation and survival, careful estimation of Tfh numbers over the entire period of the immune response and quantitative comparison of cell death and survival rates would be useful.*

We have found that survival of adoptively transferred OT2 T cells in draining LNs following OVA immunization was not affected by Raptor deficiency but was impaired in the absence of Rictor. We have added these new data in Figure 6. Additionally, we have found that expression of AktS473D can improve survival of Rictor deficient CD4 T cells ( Figure 8—figure supplement 1).

*6) The final concern relates to the mechanisms by which Rictor mediates Tfh differentiation. Additional data would greatly help clarify these. Thus, to clarify the role of Akt/GSK3b/Wnt/b-catenin/TCF1 pathway, does AktS473D over-expression in Rictor-deficient OT2 cells rescue Tcf7 and Ascl2 expression? To clarify the role of the Tcf1 p45 loop, does Tcf1 p45 loop over-expression in Rictor-deficient OT2 cells induce Bcl6 expression? The finding that forced expression of a constitutively active Akt (AktS473D) restores Bcl6 expression may indicate (as the authors speculate) that mTorc2 regulates Tfh differentiation in an Akt-independent mechanism, it is also possible that the β-catenin/Tcf1 pathway contributes to Tfh differentiation independent of the Akt-Bcl6 axis. Experiments examining whether AktS473D expression in Rictor-deficient T cells rescues the expression of β-catenin as well as Tcf1, and whether over-expression of Tcf1 lead to Bcl6 restoration would be useful to address this issue.*

We have found that expression of AKTS473D increased TCF1 protein expression and expression of TCF1 increased Bcl6 expression in adoptive transferred Rictorf/f-CD4Cre-OTII T cells (new Figure 8). We could not measure Ascl2 expression due to quality of the antibody. We regret that our experiments with TCF1-loop mutant failed due to technique reasons and that we currently do not have additional Rictorf/f-CD4Cre OTII mice for experiments. Generation of these mice requires another round of breeding. We hope the reviewers and editors would agree with us that we can address this important issue in future studies.